# Draft Genome of Tanacetum Coccineum: Genomic Comparison of Closely Related Tanacetum-Family Plants

**DOI:** 10.3390/ijms23137039

**Published:** 2022-06-24

**Authors:** Takanori Yamashiro, Akira Shiraishi, Koji Nakayama, Honoo Satake

**Affiliations:** 1Dainihon Jochugiku Co., Ltd., 1-1-11 Daikoku-cho, Toyonaka, Osaka 561-0827, Japan; t.yamashiro@kincho.co.jp (T.Y.); k.nakayama@kincho.co.jp (K.N.); 2Department of Chemical Science and Engineering, Graduate School of Engineering, Kobe University, 1-1 Rokkodai-cho, Nada-ku, Kobe, Hyogo 657-8501, Japan; 3Bioorganic Research Institute, Suntory Foundation for Life Sciences, 8-1-1 Seikadai, Seika-cho, Souraku, Kyoto 619-0284, Japan; shiraishi@sunbor.or.jp

**Keywords:** *Tanacetum coccineum*, draft genome

## Abstract

The plant *Tanacetum coccineum* (painted daisy) is closely related to *Tanacetum cinerariifolium* (pyrethrum daisy). However, *T. cinerariifolium* produces large amounts of pyrethrins, a class of natural insecticides, whereas *T. coccineum* produces much smaller amounts of these compounds. Thus, comparative genomic analysis is expected to contribute a great deal to investigating the differences in biological defense systems, including pyrethrin biosynthesis. Here, we elucidated the 9.4 Gb draft genome of *T. coccineum*, consisting of 2,836,647 scaffolds and 103,680 genes. Comparative analyses of the draft genome of *T. coccineum* and that of *T. cinerariifolium*, generated in our previous study, revealed distinct features of *T. coccineum* genes. While the *T. coccineum* genome contains more numerous ribosome-inactivating protein (RIP)-encoding genes, the number of higher-toxicity type-II RIP-encoding genes is larger in *T. cinerariifolium*. Furthermore, the number of histidine kinases encoded by the *T. coccineum* genome is smaller than that of *T. cinerariifolium*, suggesting a biological correlation with pyrethrin biosynthesis. Moreover, the flanking regions of pyrethrin biosynthesis-related genes are also distinct between these two plants. These results provide clues to the elucidation of species-specific biodefense systems, including the regulatory mechanisms underlying pyrethrin production.

## 1. Introduction

Several plants of the genus *Tanacetum* biosynthesize plant-specific metabolites, called pyrethrins, that are neurotoxic to insects. Due to their highly selective toxicity to insects and ready decomposition in the presence of sunlight and oxygen, pyrethrins are widely used as commercial insecticides in products such as mosquito coils. Pyrethrins comprise six ester compounds: pyrethrin I and II, jasmolin I and II, and cinerin I and II. One species of *Tanacetum*, *T. cinerariifolium*, has been shown to produce pyrethrins mainly in its ovary glandular trichomes, and a portion of the biosynthetic pathway for pyrethrins has been elucidated [1].

Previously, we elucidated the draft genome of *T. cinerariifolium*, the high pyrethrin-producing species [2]. *T. cinerariifolium* is known to upregulate pyrethrin biosynthesis by emitting a specific combination of volatile organic compounds (VOCs) when the plant body is wounded [3]. Interestingly, our genome analysis demonstrated that *T. cinerariifolium* encodes multiple histidine kinases, including gas-induced-type histidine kinases related to ethylene receptors, suggesting a functional correlation between histidine kinases and pyrethrin biosynthesis [2]. Furthermore, *T. cinerariifolium* possesses a greater number of ribosome-inactivating protein (RIP)-encoding genes than other plants—the apparent result of gene multiplication. In contrast, the number of genes encoding endochitinases, which are involved in the defense against fungal pathogens, is smaller than that of other plants. These features of defense-related genes suggest that *T. cinerariifolium* has undergone specific evolutionary adaptation for protection against natural predators in the dry environment of this plant’s area of origin, Dalmatia [2].

Another member of the same genus, *T. coccineum*, produces considerably lower amounts of pyrethrins, despite phylogenetically high relatedness to *T. cinerariifolium* [4] (Figure 1 and Appendix A). Moreover, *T. coccineum* is native to a region, extending from Persia to the Caucasus, which is hot and humid. In contrast, *T. cinerariifolium* originated from Dalmatia, a region that is relatively dry. Such climatic differences suggest that these two plant species may have acquired distinct defense strategies against their natural enemies that vary according to the environment. A comparison between the genomes of these two species is expected to reveal the evolutionary processes and diversification of defense strategies and to contribute to the development of technologies for the more efficient production of pyrethrins.

In the present study, we elucidated the draft genome of *T. coccineum* and performed comparative genomic analyses between these two related species. 

## 2. Results and Discussion

### 2.1. Sequencing of the T. coccineum Genome

Paired-end (PE) libraries and mate-pair (MP) libraries (with 3, 5, and 8 kb insert sizes; MP-3kb, MP-5kb, and MP-8kb, respectively) of the *T. coccineum* genome were generated and then sequenced using Illumina NovaSeq 6000 instruments. Miseq (MS) libraries were generated and then sequenced using the Illumina MiSeq system. For long reads, a PacBio (PB) library was generated and sequenced using the PacBio Sequel II system. The total base-counts of the sequence reads for PE, MP-3kb, MP-5kb, MP-8kb, MS, and PB amounted to 854 Gb, 99 Gb, 108 Gb, 109 Gb, 27 Gb, and 93 Gb, respectively (Table 1)

### 2.2. Size Estimation of the T. coccineum Genome

Prior to genomic DNA assembly, we estimated the 1C DNA content of *T. coccineum* by flow cytometry, using the *Chrysanthemum seticuspe* genome (cultivar: Gojo-0, 3 pg/1C) [5] as an internal standard. The estimated *T. coccineum* size was 9.4 pg/1C (Appendix A), which corresponds to 9.4 Gb. This size is approximately 1.6 times larger than the 5.8 pg/1C obtained in a previous report in which the size was measured by Feulgen densitometry genome estimation [6]. To further validate the 9.4 Gb genome size, we performed a k-mer spectrogram analysis of the Illumina short reads of the *T. coccineum* genome (Appendix A) using Jellyfish [7]. The k-mer spectrograms showed two major distributions: one with a maximum coverage of 1 and the other with a multimodal distribution with a maximum coverage of 44. Since the distribution with the maximum coverage = 1 was considered to be caused by sequencing errors, we treated the data after coverage = 11, which is the minimum value between the two distributions, as the k-mer derived from the correct genome. In accordance with previous studies [8], the genome size estimated from the k-mer distribution with a maximum of 44 for coverage = 11 and above was 9.8 Gb. These analyses led to the conclusion that the *T. coccineum* used in the present study has a genome size of approximately 9 Gb. In addition to the peak at Coverage = 44, an extra peak was observed at Coverage = 133 (Appendix A). Repeat sequences such as transposable elements (TEs) and simple sequence repeats (SSRs) in genomic sequences are known to evoke a multimodal distribution of k-mer spectra [9]. Since *T. cinerariifolium* is closely related to *T. coccineum* and contains many TEs in its genome sequence, this multimodal distribution suggests the proportion of TEs in the *T. coccineum* genome.

### 2.3. Sequence Assembly and Annotation of the T. coccineum Genome

The reads obtained from next-generation sequencing (NGS) were subjected to contig assembly and scaffolding in order to estimate the genome sequence (Figure 2). A collection of 6,500,576 contigs with a total length of 8.57 Gb (Table 2) was constructed by assembling PE reads and MS reads using SOAPdenovo [10]. The assembled contigs were then scaffolded with PB and MP reads using SSPACE [11,12], which concatenates contig sequences, and the Gapfiller [13] and TGS-Gapcloser programs [14], which fill the inter-contig unknown bases (‘N’s) in scaffolds with ‘A/T/G/C’ (Figure 2), as described in our previous report. Since the accuracy of gap-filled sequences depends on PB reads that exhibit a lower sequence accuracy than PE reads, the scaffold sequences were polished using POLCA [15]. The total length of the resultant scaffolds was 9.46 Gb, which corresponded well with the flow cytometry-estimated genome size for *T. coccineum* (Appendix A). The N50 value of the scaffolds was 27.8 Kb, and the maximum length of the scaffolds was 331 Kb (Table 2). Subsequently, the draft genome was subjected to analysis by AUGUSTUS [16], resulting in the prediction of 1,582,136 putative genes. These predicted genes contain unfunctional genes such as transposable elements (TEs). Compared with *T. cinerariifolium*, the value of 167,245 predicted genes/Gb in the *T. coccineum* genome was larger than the 131,830 predicted genes/Gb observed in the *T. cinerariifolium* genome [2].

The completeness of the draft genome sequences was evaluated using BUSCO [17], which counts complete (C), fragmented (F), and missing (M) conserved genes in genome sequences. The sequence analysis of 1614 conserved core plant genes confirmed that 97.8% of the conserved genes (92.7% as complete and 5.1% as fragmented) were present in the *T. coccineum* genome assembly (Table 3). These scores indicated quality as high as that obtained for *T. cinerariifolium* [2], and we therefore used the *T. coccineum* draft genome for subsequent analysis. 

Since a larger number of TEs is detected than that of functional genes in the genomes of plants, including *T. cinerariifolium*, we first analyzed the TE component of the assembled *T. coccineum* genome. TEs were detected and annotated using hmmpfam against the Gypsy database (GyDB) [18], according to the previous study [2], revealing the presence of 772,794 TEs. The value of 82,212 TEs/Gb in the *T. coccineum* genome was slightly larger than the 73,957 TEs/Gb observed in the *T. cinerariifolium* genome [2]. Furthermore, the predicted genes were subjected to InterProScan [19] to provide high-confidence annotation, revealing the presence of 103,680 putative genes encoding products that exhibited known protein signatures. Thus, a high-quality 9.4 Gb *T. coccineum* draft genome was assembled and shown to include a total of 772,794 TEs and 103,680 plausible genes based on 854 Gb of PE reads, 316 Gb of MP reads, 26.5 Gb of MS reads, and 93.1 Gb of PB reads.

### 2.4. Inter-Genus Comparative Analysis of TE Classification

We divided annotated TEs into different TE clades based on the GyDB classification and analyzed the ratio of each TE clade against all TE regions in the genomes of *T. coccineum*, *T. cinerariifolium*, *C. seticuspe*, *Artemisia annua*, *Helianthus annuus*, *Nicotiana tabacum*, *Oryza sativa*, and *Arabidopsis thaliana*. *T. cinerariifolium*, *C. seticuspe*, *A. annua*, and *H. annuus* (which belong to the Asteraceae family) and *N. tabacum*, *O. sativa*, and *A. thaliana* (which are model organism) were used as described in the previous study [2]. The top five clade ratios of each plant are shown in Table 4. In *T. coccineum*, members of the *sire-*clade TEs were the most abundant TE clade, which was also observed in the three Asteraceae plants (*T. cinerariifolium*, *C. seticuspe*, and *A. annua*). In the *T. cinerariifolium* genome, the second-largest ratio of clades was *athila*, followed (in order) by *del*, *oryco*, and *lentiviridae*; in the *T. coccineum* genome, the second-largest clade was *del*, followed (in order) by *athila*, *oryco*, and *tork*. The *T. coccineum*-specific multiplication of *del-* and *tork-*clade TEs suggested that these TEs multiplied after the evolutionary divergence of *T. cinerariifolium* and *T. coccineum* from a common ancestor.

To examine whether *del*- and *tork*-clade TEs had multiplied in a common ancestor of the Asteraceae or independently in individual genera, we estimated molecular phylogenetic trees of the reverse transcriptase (RT) domains encoded by the *del* and *tork* sequences and evaluated the number of co-clustered genes in single-genus clusters, as described in the previous study [2]. In these molecular phylogenetic analyses, the TEs that multiplied in a common ancestor are positioned in orthologous clusters, while the TEs that multiplied after divergence from a common ancestor are positioned in clusters with the TEs from single plant species (multiplied clusters). The phylogenetic analysis revealed that 67%, 62%, 73%, 68%, and 86% of the *del* TEs constituted multiplied clusters for *T. coccineum*, *T. cinerariifolium*, *C. seticuspe*, *A. annua*, and *H. annuus*, respectively (Figure 3A). Likewise, 57%, 37%, 54%, 38%, and 71% of the *tork* TEs were shown to be multiplied in the respective organisms (Figure 3B). These results indicated that more than half of the *del* TEs and more than one-third of the *tork* TEs were multiplied in the individual genera, but the other TEs were conserved as common ancestor TEs within Asteraceae. Collectively, these results suggested that most of the *del* and *tork* TEs were multiplied in the individual lineages of the respective Asteraceae genera, leading to the major TEs in *T. coccineum*.

### 2.5. Pyrethrin-Related Enzymes Encoded in the T. coccineum Genome

*T. coccineum*-predicted proteins with a high sequence similarity to nine known *T. cinerariifolium* pyrethrin biosynthesis-related proteins (TciADH2 [20], TciALDH1 [20], TciCCH [21], TciCCMT [21], TciCDS [22], TciGLIP [23], TciJMH [24], TciLOX1 [25], TciPYS [26]) were detected by BLASTP [27] (Table 5), indicating that a complete set of known pyrethrin-related enzymes is conserved in the *T. coccineum* genome. The conserved domain of each protein was detected by using InterProScan [19], indicating that these proteins were functional with a high confidence (Appendix A).

### 2.6. Synteny Analysis of Genes Encoding Pyrethrin-Related Enzymes

The distribution of genes within the scaffolds that included loci-encoding proteins corresponding to TciADH2, TciALDH1, TciCCH, TciCCMT, TciCDS, TciGLIP, TciJMH, TciLOX1, and TciPYS were analyzed using the GenomeJack software program. All of the genes were located on separate scaffolds, and the TEs were located on the flanking regions of the genes encoding all pyrethrin-related enzymes, with the exception of the genes encoding TcoCCMT and TcoGLIP. The gene encoding the Tco_1190812 protein, which contains a Jacalin-like lectin domain, was located upstream of the gene encoding the TcoCCMT protein (Figure 4A). A BLASTP search using the Tco_1190812 sequence as a query detected a predicted protein with sequence similarity (E-value of 3 × 10^−93^; 92.72% identity) to a segment of an *Artemisia annua* mannose-binding lectin (accession No. PWA73033.1). In the *T. cinerariifolium* genome, a gene encoding a corresponding Jacalin-like lectin (accession No. GEW32189.1) was also found upstream of the locus encoding TciCCMT, suggesting that this synteny is conserved.

TciGLIP (i.e., the *T. cinerariifolium* GDSL (Gly-Asp-Ser-Leu motif) lipase) is the key enzyme in the final esterification of pyrethrin biosynthesis [23]. Syntenic analysis showed that open reading frames (ORFs) encoding putative GLIPs are present in the regions downstream of both the TciGLIP- and TcoGLIP-encoding genes. However, the *T. cinerariifolium* glutathione S-transferase-encoding gene (accession No. GEU71427.1) positioned upstream of TciGLIP and the hypothetical protein-coding gene positioned downstream of tandem GLIP-encoding genes in *T. cinerariifolium* are replaced by TEs in the *T. coccineum* genome (Figure 4B), suggesting that this tandem GLIP-encoding locus translocated after the divergence of *T. coccineum* and *T. cinerariifolium*. Although the transcriptional regulatory mechanism of the TciGLIP gene is yet to be determined, this difference in the flanking region of these GLIP-encoding genes provides a clue to investigating the mechanisms regulating the possible differential expression of the genes encoding TciGLIP and TcoGLIP.

### 2.7. Functional Annotation of the T. coccineum Genes and Inter-Genus Comparative Analysis

Next, we investigated the multiplication ratios of the protein superfamilies in *T. coccineum* compared with those in other species, including *T. cinerariifolium*, which was described in our previous study. In brief, the predicted protein datasets of *T. coccineum*, *T. cinerafiiolium*, *C. seticuspe*, *A. annua*, *H. annuus*, *N. tabacum*, *O. sativa*, and *A. thaliana* were subjected to analysis using InterProScan, and multiplication odds scores were calculated for each superfamily. A positive value for the multiplication odds score indicates that a genus possesses a higher number of multiplied genes in a given superfamily than other genera.

The highest and lowest multiplication odds scores for the biodefense-, signaling-, and metabolism-related of *T. coccieum* were compared with those of other plants, including *T. cinerariifolium* (Table 6 and Table 7, respectively). To further compare *T. coccineum* with *T. cinerariifolium*, the multiplication odds scores for the superfamilies that were identified in the previous study [2] and not listed in the highest or the lowest table (Table 6 and Table 7, respectively) are shown in Table 8.

Among biodefense-related superfamilies, genes encoding proteins with the “Ribosome-inactivating protein (RIP)” (IPR036041) domain showed multiplication in the *T. coccineum* genome, exhibiting a multiplication score of 1.96 (Table 6). In the previous study, this superfamily was also multiplied in *T. cinerariifolium*. Although the odds score of the “Ribosome-inactivating protein (RIP)” (IPR036041) in the *T. coccineum* genome is 1.5 times higher than that in *T. cinerariifolium* (Table 6), the odds score of “Ricin B-like lectins” (IPR035992) in the *T. coccineum* genome is less than that of *T. cinerariifolium* (Table 8). RIPs, including ricin, show high toxicity to a wide range of species, including insects, bacteria, and viruses, serving as biodefense molecules for the producing plant [28]. RIPs are categorized into type I and type II due to the absence or presence (respectively) of the ricin B lectin domain [29]. The ricin B lectin domain is involved in internalization via the binding of target cell glycans; therefore, type-II RIPs have a higher toxicity than type-I RIPs. Taken together, these results demonstrated that genes encoding higher-toxicity type-II RIPs are multiplied (i.e., more abundant) in the *T. cinerariifolium* genome compared with those in the *T. coccineum* genome, suggesting that *T. coccineum* may have been subjected to, or may be more sensitive to, natural enemies under wild conditions compared with *T. cinerariifolium*. In the previous study, a gene encoding a putative insecticidal type-II RIP Tci_399175 (accession No. GEY27201.1) that showed sequence similarity to the *Sambucus nigra* insecticidal RIP SNA-I (*S. nigra* agglutinin-I, accession No. O22415.1) [30] was found in the *T. cinerariifolium* genome [2]. A BLASTP search of the *T. coccineum* genome with the SNA-I sequence returned Tco_1336120. An alignment of the RICIN domain, which is important for identifying target cells, confirmed that this putative insecticidal RIP is also encoded in the *T. coccineum* genome (Appendix A). These results indicated that an SNA-I-like insecticidal RIP is conserved in both *Tanacetum* species. In combination, these comparative analyses of the RIP genes verified that type-I and type-II RIPs are abundant in the *T. coccineum* and *T. cinerariifolium* genomes, respectively, suggesting distinct RIP-associated defense strategies between these two plants.

The “Endochitinase-like superfamily” (IPR036861), which plays a pivotal role in the defense against fungal pathogens, is present in the *T. coccineum* genome at levels similar to those seen in other genera, and it is more abundant than that seen in *T. cinerariifolium* (Table 8). While *T. cinerariifolium* is native to a region with a dry environment, *T. coccineum* is native to a region with a humid environment, indicating that these plants have been exposed to distinct natural enemies. These observations suggested that distinct defense strategies may have evolved in the different lineages leading to *T. cinerariifolium* and *T. coccineum*, reflecting the differing areas of origin of the two species.

The metabolism-related superfamily showed the highest gene multiplication values in *T. coccineum*, with genes encoding “Urease, alpha subunit” (IPR005848), “Lipoxygenase, C-terminal domain” (IPR036226), “RuBisCo” (IPR033966), “Metal-dependent hydrolase” (IPR032466), “Enolase-like, C-terminal domain” (IPR036849), and “Cytochrome P450” (IPR036396) exhibiting multiplication scores of 2.36, 1.86, 1.60, 1.48, 1.38, and 0.90, respectively (Table 6). In particular, genes encoding metalloproteins such as lipoxygenases, metal-dependent hydrolases, and cytochrome P450 were multiplied, which was also observed for the *T. cinerariifolium* genome [2]. Since some pyrethrin-related proteins belong to the superfamily of cytochrome P450s or lipoxygenases, the corresponding genes may have multiplied in the common lineage shared by *T. coccineum* and *T. cinerariifolium*, given that both species have the ability to synthesize pyrethrins. The cytochrome P450 superfamily-encoding genes were 1.6 times more numerous in the *T. coccineum* genome than in the *T. cinerariifolium* genome. Molecular phylogenetic analysis showed that 57% of the *T. coccineum* cytochrome P450s were not included in orthologous gene clusters but constituted single-genus clusters (Figure 5), suggesting that some of the orthologous cytochrome P450s were multiplied in the *T. coccineum*-specific lineage. These results support the view that the cytochrome P450s of *T. cinerariifolium* and *T. coccineum* might have multiplied during the respective evolutionary processes and then acquired an ability to produce species-specific plant specialized metabolites, including pyrethrins. Moreover, these results suggested that species-specific secondary metabolites may be more abundant in *T. coccineum* than in *T. cinerariifolium*. The further investigation of *T. coccineum* secondary metabolites is needed.

While the high multiplication of proteins harboring the “HECT, E3 ubiquitin ligase catalytic domain” (IPR035983) was observed in *T. coccineum* (Table 6) (as is the case in *T. cinerariifolium*), low multiplication was seen for proteins containing “RCHY, zinc-ribbon” (IPR039512), a domain that is contained in the RING-finger-type E3 ubiquitin ligases (Table 7). These results suggested that genes encoding the HECT-type E3 ubiquitin ligases are multiplied in the *T. coccineum* genome, while genes encoding the RING-finger-type E3 ubiquitin ligases are not. The investigation of the biological significance of this apparent imbalance in the amplification of E3 ubiquitin ligase-encoding genes awaits further study. 

Similarly, genes encoding proteins containing the “Signal transduction histidine kinase, dimerization/phosphoacceptor domain” (IPR036097) are multiplied in *T. cinerariifolium* but not in the *T. coccineum* genome (Table 8). In planta, histidine kinases are involved in responding to environmental stimuli, including sunlight, plant hormones, and ethylene. *A. thaliana* ethylene receptor 1 (AtETR1) is a typical gas-induced histidine kinase that possesses both a HATPase (histidine kinase-like ATPase) domain and an REC (phosphoacceptor receiver) domain [31]. We surveyed the histidine kinase-encoding genes of *T. coccineum* and *T. cinerariifolium* for the presence of the HATPase and REC domains; the data for the number of genes containing each domain are presented as Venn diagrams in Figure 6A. The number of genes encoding both the HATPase and REC domains were 13 and 38 for *T. coccineum* and *T. cinerariifolium*, respectively. A molecular phylogenetic tree of the predicted histidine kinase proteins is presented in Figure 6B; the AtETR1 protein is indicated, as are clusters for the 5 and 4 paralogs found in *T. cinerariifolium* and *T. coccineum*, respectively. The comparative analysis detected not only orthologous clusters but also the apparent multiplication of a *T. cinerariifolium*-specific single-genus cluster (Figure 6B, green). 

Our previous study implicated a gas-induced histidine kinase in the VOC-mediated regulation of pyrethrin production in *T. cinerariifolium* [2]. The present study also suggested a correlation between the extent of pyrethrin production and the number of histidine kinase proteins in *T. coccineum* and *T. cinerariifolium*. These results suggested that *T. coccineum* has not acquired a gas (i.e., VOC)-induced pyrethrin production system via the species-specific multiplication of histidine kinase-encoding genes distinct from *T. cineariifolium*. An investigation of the functional relationship between histidine kinases and VOC-induced pyrethrin production is underway. 

In this study, we elucidated the draft genome of *T. coccineum*. Comparative genomic analyses between *T. coccineum* and a closely related species, *T. cinerariifolium*, revealed characteristic features of *T. coccineum* genes, leading to the difference in pyrethrin production between *T. coccineum* and *T. cinerariifolium*.

## 3. Materials and Methods

### 3.1. Phylogenetic Analysis of Plants in This Paper

The sequences of the internal transcribed spacer (ITS)1, ITS2, ribulose-1,5-bisphosphate carboxylase/oxygenase large subunit (*rbcL*), and maturase K (*matK*) were obtained from the NCBI database or BLASTN. Accession No.; *T*. *cinerariifolium*: AB359720.1 (ITS1), AB359806.1 (ITS2), and MT104464.1 (*rbcL* and *matK*); *T*. *coccineum*: AB359721.1 (ITS1), AB359807.1 (ITS2), and MT104463.1 (*rbcL* and *matK*); *A*. *annua*: KC493085.1 (ITS1 and ITS2) and MF623173.1 (*rbcL* and *matK*); *H*. *annuus*: KF767534.1 (ITS1 and ITS2), L13929.1 (*rbcL*), and AY215805.1 (*matK*); *N. tabacum*: AJ300215.1 (ITS1 and ITS2), AP019625.1 (*rbcL*), and MZ707522.1 (*matK*); *A. thaliana*: X52320.1 (ITS1 and ITS2), NC_000932.1 (*rbcL*), and MK380721.1 (*matK*); *O. sativa*: KM036282.1 (ITS1 and ITS2), D00207.1 (*rbcL*), and KM103369.1 (*matK*). The sequences of the ITS 1, ITS2, *rbcL*, and *matK* of *C. seticuspe* were detected by BLASTN (version 2.7.1) [27], with the sequences of these genes of *A. thaliana* as queries. The nucleic acid sequences (except *O. sativa* sequences due to their low similarity to other plants) were aligned using CLUSTAL W-mpi 0.13 [32], and maximum-likelihood phylogenetic trees based on a JTT matrix-based model [33] with 1000 bootstraps were created using the IQ-TREE 2.0.3 [34] and FigTree v1.4.4 software (http://tree.bio.ed.ac.uk/software/figtree/ (accessed on 27 May 2022)).

### 3.2. Plant Materials and Genome Sequencing

The seeds of *T. coccineum* (cultivar: Robinson mix) were obtained commercially from Sakata Seed Co., Ltd., Japan (production number, 906435). Genomic DNA was extracted from the seeds using a DNeasy Plant Mini Kit (Qiagen), according to the manufacturer’s instructions. PE and MP libraries with three different insert sizes (3, 5, and 8 kb) of the extracted DNA were constructed using a TruSeq DNA PCR-Free kit (Illumina) and a Nextera Mate-Pair Sample Prep Kit (Illumina), respectively. The PE and MP libraries were then subjected to 151 × 2 cycles of paired-end sequencing, using NovaSeq 6000 Illumina instruments. MS libraries of the extracted DNA were constructed using a TrueSeq DNA PCR-Free kit (Illumina). The MS library was subjected to 301 × 2 cycles of paired-end sequencing using an Illumina Miseq System. PB libraries of the extracted DNA were constructed using a SMRTbell Express Template Prep Kit (PacBio). The PB libraries were subjected to sequencing using a PacBio Sequel II system.

### 3.3. Genome Size Estimation Using Flow Cytometry

The *T. coccineum* was grown from seed under field conditions for 6 months, and a leaf was obtained on the resulting plants. A 5 mm square section was excised from the leaf using a razor and treated with Quantum Stain UV and PI for DNA (Quantum Analysis), according to the manufacturer’s protocol. The genome size of *T. coccineum* was estimated using a CyFlow SL flow cytometer (Sysmex Partec). Segments from the leaves of *T. cinerariifolium* (wild type, 7.1 Gb) and *C. seticuspe* (3 Gb, cultivar: Gojo-0) [5] were also assayed as size references. The *C. seticuspe* Gojo-0 strain used in this study and related information are available from the National BioResource Project (https://shigen.nig.ac.jp/chrysanthemum/top.jsp (accessed on 26 October 2020)).

### 3.4. Genome Size Estimation Using K-Mer Depth Information

Separately, the genome size was estimated using k-mer depth analysis, as described previously [8]. Briefly, the occurrence of each k-mer was counted using the Jellyfish software [7], and the homozygous k-mer depth peak (*C*) was determined from a histogram of the k-mer depths. The genome size was then calculated using the formula
$$Genome\;size=\frac{n*\left(L-k+1\right)}{C}$$
where *n, L*, and *k* indicate the total number of reads, the average read length, and the k-mer size, respectively

### 3.5. De Novo Assembly of Genome Sequences

The obtained read sequences were cleaned for assembly as described in our previous study [2]. Adapters derived from the Truseq or Nextera mate-pair Sample Prep Kit, low-quality reads, and short reads (<36 bp in length) were trimmed using Trimmomatic version 0.36 [35].

Contig sequences were generated from the PE reads by a four-step process (“pre-assembly”, “SOAPdenovo assembly”, “clean-up”, and “merging with other assembly result”). First, the cleaned PE reads, including overlaps, were detected and pre-assembled with PANDAseq [36]. Second, these pre-assembled reads and remaining reads were subjected to SOAPdenovo v2.04-r240 [10] with multi k-mers from 80 to 127 to generate contig sequences. Third, the SOAPdenovo-generated contigs were cleaned by trimming and merging the contigs showing more than 95% sequence identity with other contig sequences. 

In contrast to the previous study, a “hybrid assembling” process was incorporated to generate pre-scaffolds using SSPACE-longread v1-1 [11], with PacBio reads as guides, including setting the minimum overlap length to 20 bp, the minimum number of links to 3, and the maximum link ratio to 0.3.

Scaffold sequences were generated by a three-step process (“scaffolding”, “gapfilling”, and “polishing”). First, PE and MP reads were mapped to contig sequences using bowtie version 2.3.4.3 [37] with the --local option; read pairs mapped discordantly were subjected to the following steps, and the selected reads and contigs were scaffolded using SSPACE-STANDARD version 3.0 [12] (BaseClear), including setting the minimum number of links to 3. Second, the positions for which the bases were unknown (e.g., ‘N’) in the scaffolds were filled by GapFiller v1-10 [13] (BaseClear), for which the minimum number of overlaps was set to 30 and the number of reads to trim off was set to 10, and then filled by using TGS-Gapcloser [14] with the default parameter. Additionally, the scaffolds were polished using the POLCA program [15] to complete the draft genome. Third, the completeness of the draft genome was evaluated using BUSCO-v5 [38], with the embryophyta_odb9 protein set.

### 3.6. Gene Prediction and Annotation

The assembled genome was subjected to analysis with AUGUSTUS 3.3.1 [16], trained with the training data generated by BUSCO using the --long option. The TEs in the predicted genes were identified by hmmpfam in HMMER 2.3.1 [39] and by a comparison to GyDB 2.0 [18] using an E-value cutoff of 1.0, according to the previous reports [2,40]. To identify high-confidence genes, the genes encoding proteins with known protein signatures were detected using InterProScan 5.33–72.0 [19] and annotated using Blast2GO [41].

### 3.7. Comparative Analysis of TE Content Versus That in Other Plants

The TE content was also estimated for six other genome-elucidated plants, including *A. thaliana* (TAIR10 [42]), *N. tabacum* (Ntab-TN90 [43]), *O. sativa* (assembly Build 4.0 [44]), *H. annuus* (HA412HO_v1.1 [45]), *A. annua* (ASM311234v1 [46]), and *C. seticuspe* (CSE_r1.0 [40]). As described for the TE detection in *T. cinerariifolium*, the coding regions of these genomes were estimated using AUGUSTUS 3.3.1 [16] with the Arabidopsis model set and the default parameters, and the TEs in these predicted transcriptomes were extracted using hmmpfam and a comparison to GyDB. According to GyDB classifications, the percentage of genomic regions occupied by each clade of TEs was calculated as an accumulation score. 

Molecular phylogenetic trees for the *del-* and *tork-*clade TEs also were estimated using the ORTHOSCOPE method [47], as described in our previous study [48]. The amino acid sequences of hmmpfam-extracted reverse-transcriptase domains encoded by the TEs in *T. cinerariifolium*, *H. annuus*, *A. annua*, and *C. seticuspe* were aligned using CLUSTAL W-mpi 0.13 [32], and maximum-likelihood phylogenetic trees based on a JTT matrix-based model [33] with 100 bootstraps were created using Fast Tree 2.1.10 (JTT model, CAT approximation) [49].

### 3.8. Homology Search and Synteny Analysis of Genes Encoding Pyrethrin-Related Enzymes

*T. coccineum*-predicted proteins with high homology to known pyrethrin-related enzymes were detected by BLASTP (version 2.7.1) [27], with the default parameters. The top hit sequences with a greater than 85% overall identity to *T. cinerariifolium* pyrethrin biosynthesis-related enzymes—TciADH2 (accession No. AUQ44118.1), TciALDH1 (accession No. AUQ44119.1), TciCCH (accession No. AGO03787.1), TciCCMT (accession No. QCP80351.1), TciCDS (accession No. ADO17798.1), TciGLIP (accession No. AFJ04755.1), TciJMH (accession No. AXL93690.1), TciLOX1 (accession No. AGO03785.1), and TciPYS (accession No. AXL93709.1)—were regarded as *T. coccineum* pyrethrin biosynthesis-related enzymes. Each detected protein (Table 5) was subjected to sequence alignment using CLUSTAL W-mpi 0.13 [45], to domain search using InterProScan 5.33–72.0 [19], and to synteny analysis using the GenomeJack software program (https://genomejack.net/english/index.html).

### 3.9. Comparative Analysis of Protein Superfamily Content Versus That in Other Plants

We also detected protein signatures using InterProScan [19] analysis in six genome-elucidated plants in the previous study and *T. cinerariifolium*, using the methods described above for the InterProScan analysis of *T. coccineum*. Having determined the number of genes possessing each superfamily signature, the multiplication odds score for each combination of an InterProScan-detected superfamily signature (Sig) and plant genus (Genus) was calculated as follows:$$\mathrm{Multiplication}\;\mathrm{odds}\;\mathrm{score}\;\left(\mathrm{Genus},\;\mathrm{Sig}\right)=\log_{2}\frac{N\left(Genus,\;Sig\right)+PS}{\bar{N\left(Sig\right)}+PS}$$
where *N* (Genus, Sig) represents the number of the plant-genus genes with an InterProScan-detected superfamily signature and PS represents the pseudo-count constant, which was set to 5.00.

In a further analysis for functional proteins, we subjected the predicted proteins of the RIP-related superfamily to a BLASTP search (version 2.7.1) [27]. *S. nigra* agglutinin I (SNA-I, accession No. O22415.1) [30] was used as the query for RIPs. The detected proteins (SNA-I, Tci_3991752, and Tco_1336120) were subjected to sequence alignment using CLUSTAL W-mpi 0.13 [32], and maximum-likelihood phylogenetic trees based on a JTT matrix-based model [33] with 500 bootstraps were created using the MEGA software [50].

The predicted histidine kinases of *T. coccineum* and *T. cinerariifolium* were subjected to searches using the Conserved Domain Database (CDD v.3.19) [51] to detect HATPase (histidine kinase-like ATPase) and REC (phosphoacceptor receiver) domains. Molecular phylogenetic trees were further estimated for histidine kinases. The AtETR1 and histidine kinases of *T. coccineum* and *T. cinerariifolium* were aligned using CLUSTAL W-mpi 0.13 [32], and maximum-likelihood phylogenetic trees based on a JTT matrix-based model [33] with 500 bootstraps were created using the MEGA software [50].

## Figures and Tables

**Figure 1 ijms-23-07039-f001:**
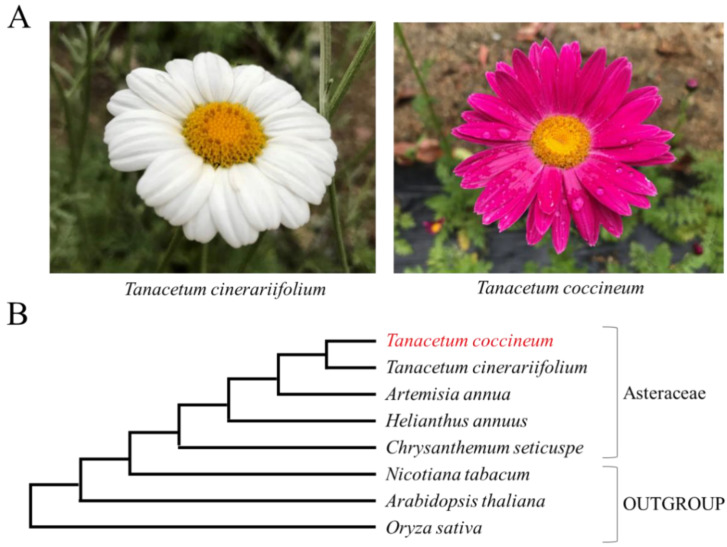
(**A**) Flowers of *Tanacetum cinerariifolium* and *Tanacetum coccineum*. (**B**) The phylogenetic tree of plants in this paper. To generate the phylogenetic tree, the sequences of the internal transcribed spacer (ITS)1, ITS2, ribulose-1,5-bisphosphate carboxylase/oxygenase large subunit (*rbcL*), and maturase K (*matK*) were obtained from the NCBI database or BLASTN search. The nucleic acid sequences of ITS1, ITS2, *rbcL*, and *matK* were aligned using CLUSTAL W-mpi 0.13, and maximum-likelihood phylogenetic trees based on a JTT matrix-based model with 1000 bootstraps were created using the IQ-TREE 2.0.3 and FigTree v1.4.4 software.

**Figure 2 ijms-23-07039-f002:**
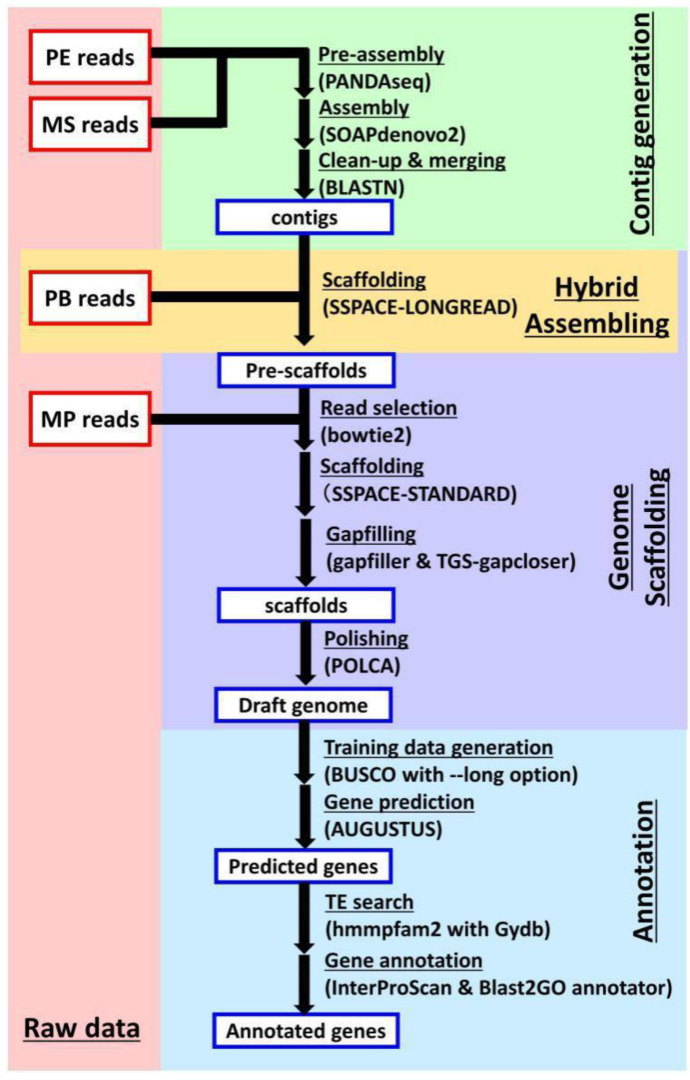
Flowchart of genome assembly and gene prediction. The paired-end (PE) reads and Miseq (MS) reads were subjected to contig assembly by a three-step process, including “pre-assembly” using PANDAseq, “contig assembly” using SOAPdenovo, and “clean-up & merging” using BLASTN. The PacBio (PB) reads were subjected to hybrid assembly using SSPACE-LONGREAD, generating pre-scaffolds. Then, the mate-pair (MP) reads were subjected to scaffold generation by a four-step process, including “read selection” using bowtie2, “scaffolding” using SSPACE-STANDARD, “gapfilling” using Gapfiller and TGS-Gapcloser, and “polishing” using POLCA. These processes yielded the complete draft genome. The coding sequences in the draft genome were then annotated by a four-step process, including “Training data generation” using BUSCO, “Gene prediction” using AUGUSTUS, “TE search” (TE, transposable element) using hmmpfam, and “Gene annotation” using InterProScan and Blast2GO.

**Figure 3 ijms-23-07039-f003:**
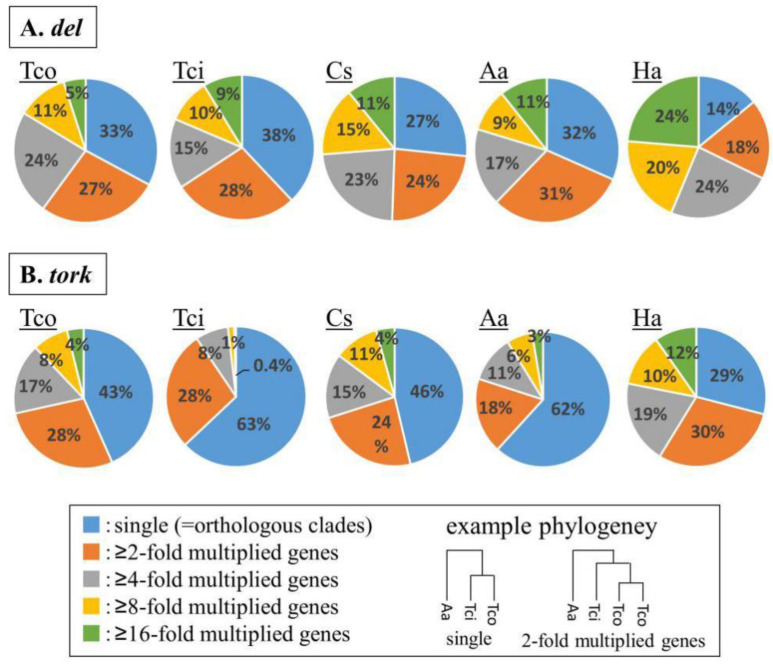
Multiplication analysis of (**A**) *del-*clade and (**B**) *tork-*clade transposable elements (TEs). Based on the molecular phylogenetic trees for *del* TEs and *tork* Tes, the number of co-clustered genes in single-genus clusters, which reflects the inferred number of genus-specific duplication events, were counted for each genus. Tco: *T. coccineum*; Tci: *T. cinerariifolium*; Cs: *C. seticuspe*; Aa: *A. annua*; Ha: *H. annuus.*

**Figure 4 ijms-23-07039-f004:**
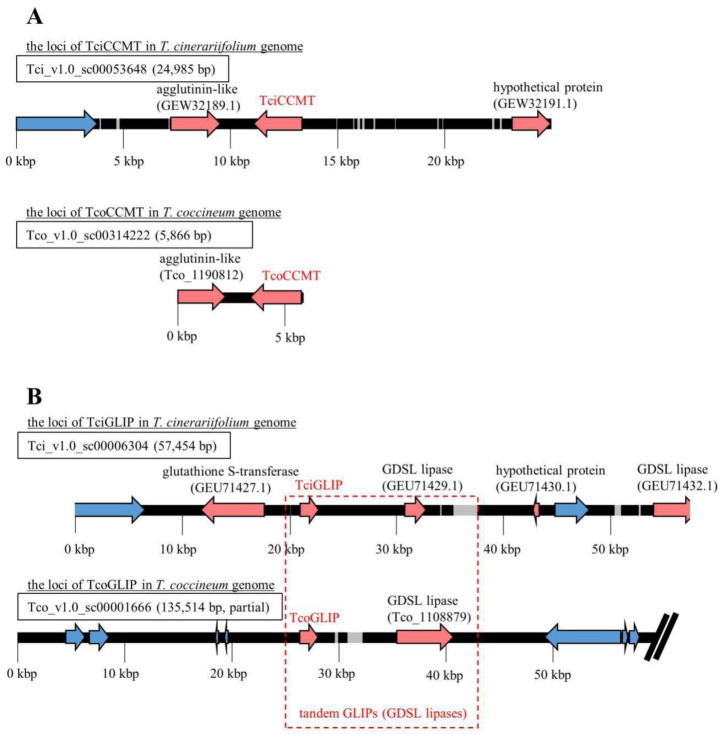
Scaffolds including genes encoding the pyrethrin-related enzymes Tci(o)CCMT (**A**) and Tci(o)GLIP (**B**). Gray region: gap; red arrows: protein-coding gene; blue arrow: transposable element (TE) gene.

**Figure 5 ijms-23-07039-f005:**
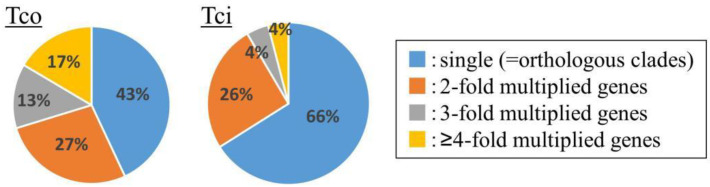
Multiplication analysis of genes encoding proteins with the superfamily “cytochrome P450” (IPR036396) domain in the *T. coccineum* and *T. cinerariifolium* genomes. Tco: *T. coccineum*; Tci: *T. cinerariifolium*.

**Figure 6 ijms-23-07039-f006:**
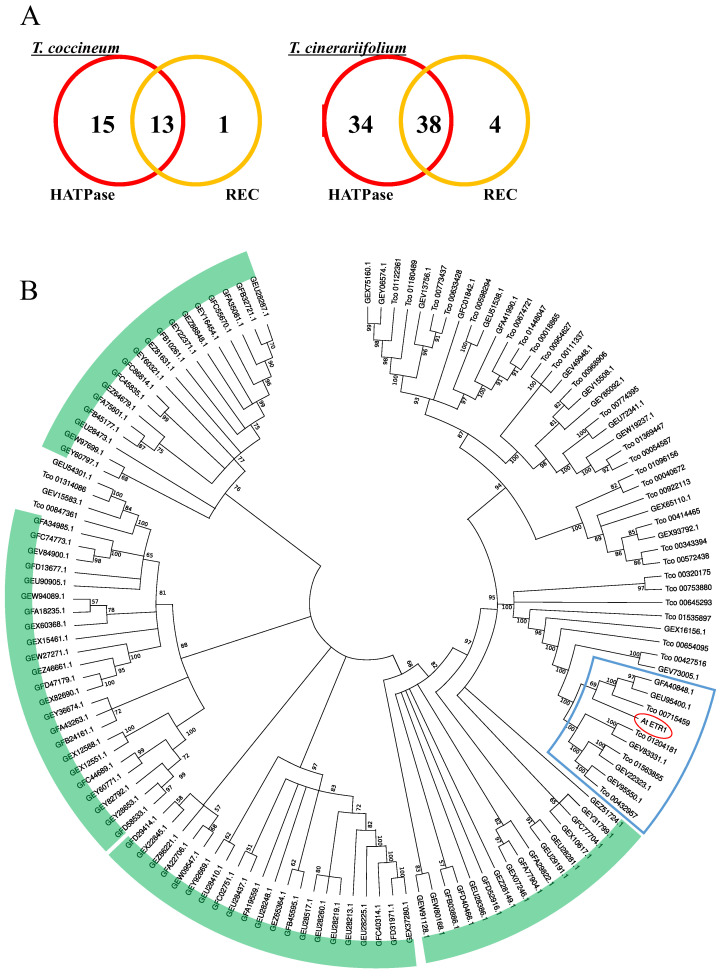
The number of gene-encoding proteins containing histidine kinase domains (**A**), and the phylogenetic analysis (**B**) of histidine kinase superfamily genes. (**A**) Venn diagram of the numbers of *T. coccineum* and *T. cinerariifolium* histidine kinase superfamily genes encoding proteins with histidine kinase-like ATPase (HATPase) or phosphoacceptor receiver (REC) domains. (**B**) A phylogenetic tree of the *T. coccineum* and *T. cinerariifolium* genes encoding proteins belonging to the signal transduction histidine kinase superfamily. The *T. coccineum* genes and *T. cinerariifolium* genes are shown as the name with the prefix “Tco” and the accession No., respectively. The *A. thaliana* ethylene receptor 1 protein (AtETR1) is circled in red; the AtETR1 cluster that includes five paralogs found in *T. cinerariifolium* and four paralogs found in *T. coccineum* is surrounded by a blue line; *T. cinerariifolium*-specific clusters are indicated by green arcs.

**Table 1 ijms-23-07039-t001:** Statistics of sequence reads.

Library	Insert Size (bp)	Read Length (Bases)	Number of Reads	Total Read Length (Bases)
PE	350	151	5,732,398,372	854,270,829,961
MP-3kb	3000	151	698,859,570	99,096,491,543
MP-5kb	5000	151	750,513,382	107,931,325,410
MP-8kb	8000	151	762,709,822	109,359,355,641
MS	550	301	97,731,712	26,503,089,921
PB		Ave. 10,738	8,670,092	93,100,193,428

Read lengths for Illumina NGS (PE, MP, MS) were indicated in max length, and PacBio read (PB) was indicated in average length. PE: paired-end; MP: mate-pair; MS: Miseq; PB: PacBio; NGS: next-generation sequencing.

**Table 2 ijms-23-07039-t002:** Statistics of genome assembly.

	Contigs	Scaffolds (before Gapfilling)	Draft Genome
Total number of sequence fragments	6,500,576	3,061,809	2,836,647
Total length (bp)	8,565,698,618	9,395,951,224	9,463,677,832
N50 (bp)	8465	25,397	27,784
Length of longest contig (bp)	149,916	329,693	331,286
Gaps (bp)	0	777,041,487	724,210,424
GC content (%)	34.9	35.1	35.1

**Table 3 ijms-23-07039-t003:** Annotation statistics for the draft genome.

Number of Predicted Genes	1,582,136
BUSCO v5	* C: 92.7% (Single: 70.8%, Duplicated: 21.9%) F: 5.1% M: 2.2%
Number of predicted TEs	772,794
Number of predicted genes encoding products with known protein signatures	103,680

* C: percentage of full-length conserved genes in BUSCO notation; F: percentage of fragmented conserved genes in BUSCO notation; M: percentage of missing genes in BUSCO notation; TE: transposable element.

**Table 4 ijms-23-07039-t004:** Annotation statistics for the draft genome.

Rank	Tco	Tci	Cs	Aa	Ha	Nt	Os	At
1	sire (25.7)	sire (33.0)	sire (32.0)	sire (21.8)	del (37.7)	del (40.4)	tat (11.4)	athila (9.54)
2	del (15.3)	athila (17.0)	athila (10.9)	athila (19.6)	sire (9.85)	tat (20.5)	retroviridae (8.97)	retroviridae (4.89)
3	athila (12.5)	del (12.0)	oryco (5.11)	del (6.57)	lentiviridae (8.72)	athila (9.87)	del (8.39)	caulimovirus (4.15)
4	oryco (7.25)	oryco (6.34)	lentiviridae (5.06)	oryco (4.59)	tat (6.76)	sire (3.02)	tork (4.73)	badnavirus (4.05)
5	tork (4.70)	lentiviridae (4.92)	del (5.03)	tork (4.01)	athila (5.17)	tork (2.80)	alpharetroviridae (4.66)	tork (3.08)

Parenthesized numbers indicate the ratio (%) of each clade against the total TE regions in that species. Tco: *T. coccineum*; Tci: *T. cinerariifolium*; Cs: *C. seticuspe*; Aa: *A. annua*; Ha: *H. annuus*; Nt: *N. tabacum*; Os: *O. sativa*; At: *A. thaliana*.

**Table 5 ijms-23-07039-t005:** *T. coccineum* genome-encoded proteins corresponding to known pyrethrin biosynthesis-related proteins.

Known Pyrethrin-Related Enzymes	Corresponding Proteins of *T. coccineum*	Protein Sequence Similarity
TciADH2	Tco_0487905	Identities = 340/378 (90%), Positives = 359/378 (95%), Gaps = 2/378 (1%)
TciALDH1	Tco_0682217	Identities = 448/499 (90%), Positives = 471/499 (94%), Gaps = 1/499 (0%)
TciCCH	Tco_0360514	Identities = 470/498 (94%), Positives = 484/498 (97%), Gaps = 1/498 (0%)
TciCCMT	Tco_1190813	Identities = 358/374 (96%), Positives = 361/374 (97%), Gaps = 5/374 (1%)
TciCDS	Tco_1315810	Identities = 358/395 (91%), Positives = 374/395 (95%), Gaps = 0/395 (0%)
TciGLIP	Tco_1108878	Identities = 337/365 (92%), Positives = 348/365 (95%), Gaps = 0/365 (0%)
TciJMH	Tco_0572988	Identities = 450/512 (88%), Positives = 479/512 (94%), Gaps = 2/512 (0%)
TciLOX1	Tco_0863779	Identities = 847/861 (98%), Positives = 853/861 (99%), Gaps = 0/861 (0%)
TciPYS	Tco_1240348	Identities = 465/488 (95%), Positives = 475/488 (97%), Gaps = 0/488 (0%)

The information of the “Protein sequence similarity” column is introduced by the BLASTP program with each known pyrethrin-related enzyme as a query. Tci: *T. cinerariifolium*; Tco: *T. coccineum*; ADH2: alcohol dehydrogenase 2; ALDH1: aldehyde dehydrogenase 1; CCMT: 10-carboxychrysanthemic acid 10-methyltransferase; CDS: chrysanthemyl diphosphate synthase; CHH: chrysanthemol 10-hydroxylase; GLIP: GDSL (Gly-Asp-Ser-Leu motif) lipase; JMH: jasmone hydroxylase; LOX1: 13-lipoxygenase; PYS: pyrethrolone synthase.

**Table 6 ijms-23-07039-t006:** Superfamilies with the highest multiplication odds scores in *T. coccineum*.

Category	IPR ID	Superfamily Name	Tco	Tci	Cs	Aa	Ha	Nt	Os	At
Biodefense	IPR036041	Ribosome-inactivating protein	1.96 (159)	1.29 (98)	−1.81 (7)	−1.00 (16)	−3.07 (0)	−3.07 (0)	−0.94 (17)	−3.07 (0)
Metabolism	IPR005848	Urease, alpha subunit	2.36 (108)	−0.14 (15)	−1.87 (1)	−1.46 (3)	−1.87 (1)	−0.87 (7)	−2.14 (0)	−1.87 (1)
Metabolism	IPR036226	Lipoxygenase, C-terminal domain	1.86 (232)	0.48 (86)	−0.22 (51)	−0.82 (32)	−1.12 (25)	−0.67 (36)	−1.86 (13)	−2.44 (7)
Metabolism	IPR033966	RuBisCO	1.60 (42)	−0.25 (8)	0.05 (11)	−0.15 (9)	−1.37 (1)	−0.95 (3)	−0.25 (8)	−1.15 (2)
Metabolism	IPR032466	Metal-dependent hydrolase	1.48 (166)	0.69 (94)	−1.13 (23)	−0.89 (28)	−0.32 (44)	−0.05 (54)	−1.61 (15)	−0.98 (26)
Metabolism	IPR036849	Enolase-like, C-terminal domain	1.38 (71)	0.91 (50)	−0.62 (14)	−0.55 (15)	−0.78 (12)	−0.29 (19)	−1.41 (6)	−1.29 (7)
Metabolism	IPR036396	Cytochrome P450	0.90 (1220)	0.19 (745)	0.16 (732)	0.07 (688)	−0.20 (568)	−0.12 (600)	−1.05 (314)	−0.85 (361)
Signaling	IPR035983	HECT, E3 ligase catalytic domain	1.22 (95)	0.84 (72)	0.10 (41)	−0.52 (25)	−0.90 (18)	−0.21 (32)	−1.84 (7)	−1.25 (13)

Parenthesized numbers indicate the number of genes categorized in each superfamily. Tco: *T. coccineum*; Tci: *T. cinerariifolium*; Cs: *C. seticuspe*; Aa: *A. annua*; Ha: *H. annuus*; Nt: *N. tabacum*; Os: *O. sativa*; At: *A. thaliana*.

**Table 7 ijms-23-07039-t007:** Superfamilies with the lowest multiplication odds scores in *T. coccineum*.

Category	IPR ID	Superfamily Name	Tco	Tci	Cs	Aa	Ha	Nt	Os	At
Signaling	IPR039512	RCHY1, zinc-ribbon	−1.18 (3)	−0.48 (8)	−0.09 (12)	−0.09 (12)	−0.01 (13)	1.28 (39)	−0.72 (6)	−0.09 (12)

Parenthesized numbers indicate the number of genes categorized in each superfamily. Tco: *T. coccineum*; Tci: *T. cinerariifolium*; Cs: *C. seticuspe*; Aa: *A. annua*; Ha: *H. annuus*; Nt: *N. tabacum*; Os: *O. sativa*; At: *A. thaliana*.

**Table 8 ijms-23-07039-t008:** Superfamilies with characteristic odds scores in the *T. cinerariifolium* genome.

Category	IPR ID	Superfamily Name	Tco	Tci	Cs	Aa	Ha	Nt	Os	At
Biodefense	IPR035992	Ricin B-like lectins	0.81 (44)	1.41 (69)	−0.34 (17)	−0.05 (22)	−1.22 (7)	−0.80 (11)	−1.10 (8)	−1.48 (5)
Biodefense	IPR036861	Endochitinase-like	−0.13 (7)	−1.13 (1)	−0.39 (5)	−0.25 (6)	0.53 (14)	0.29 (11)	0.09 (9)	0.37 (12)
Signaling	IPR036097	Signal transduction histidine kinase, dimerization/phosphoacceptor domain	−0.11 (32)	1.41 (101)	−0.62 (21)	−0.37 (26)	−0.28 (28)	0.35 (46)	−1.74 (7)	−0.74 (19)
Signaling	IPR024792	Rho GDP-dissociation inhibitor domain	0.48 (18)	1.24 (34)	−0.14 (10)	−0.58 (6)	−0.34 (8)	−0.14 (10)	−1.04 (3)	−1.04 (3)
Metabolism	IPR012347	Ferritin-like	0.72 (22)	1.29 (35)	−0.03 (11)	−0.71 (5)	−0.57 (6)	−0.86 (4)	−1.23 (2)	−0.57 (6)
Metabolism	IPR036909	Cytochrome c-like domain	0.40 (21)	1.16 (39)	−0.50 (9)	−0.84 (6)	−0.22 (12)	−0.16 (17)	−0.82 (7)	−0.82 (7)
Metabolism	IPR037069	Acyl-CoA dehydrogenase/oxidase, N-terminal domain	0.45 (22)	1.05 (36)	−0.60 (8)	−0.30 (11)	−0.22 (12)	0.22 (18)	−0.98 (5)	−0.84 (6)

Parenthesized numbers indicate the number of genes categorized in each superfamily. Tco: *T. coccineum*; Tci: *T. cinerariifolium*; Cs: *C. seticuspe*; Aa: *A. annua*; Ha: *H. annuus*; Nt: *N. tabacum*; Os: *O. sativa*; At: *A. thaliana*.

## Data Availability

The draft genome sequences and annotations, along with the raw reads for PE, MP, MS, and PB, have been deposited in the DNA Data Bank of Japan (DDBJ) under the BioProject accession Code PSUB016075.

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
