# Peer review of "Draft Genome of Tanacetum Coccineum: Genomic Comparison of Closely Related Tanacetum-Family Plants"

_ijms, 2022, doi:10.3390/ijms23137039_

Round 1
Reviewer 1 Report
Yamashiro et al, presented the first draft genome of T.coccineum. Comparing T.coccineum to intra- and inter-genus species, the authors found distinctive gene families that are specific to T.coccineum. The authors did a good job on comparative analyses, but other analyses such as gene annotation and TE analysis need to be improved.
-
Phylogenetic analysis of plants. Only ITS1 and ITS2 are not enough to draw a phylogenetic tree, especially the plants selected all have good genome assemblies. It is 2022, the authors should use multiple markers, rather than ITS alone for this type of analysis.
-
Please report the number of Ns for contigs and scaffold assemblies in Table 2. I wonder about the change of gaps after you applied the gap-filling step.
-
Did you do repeat masking before gene annotation? You should find a lot smaller gene numbers after masking repeats. I would suggest the authors re-annotate the genome.
-
The number of TEs together with the number of protein-coding genes are 876,474, the authors annotated a total of 1,582,136. What are the other half of the predicted genes?
-
Gypsy (Athila, Tat, Galadriel, Teina, Del/Tekay) and Copia (TAR/Tork, Angela/Tork, Ikeros/Tork, Maximus/Sire, Ivana/Oryco, Ale/Retrofit…) are the two big classes of TEs, most of the TEs in plant genomes are these two superfamilies, it is not necessary to present the subgroups, but lost resolutions for the lower amount TEs. I recommend the authors use RepeatMasker to reanalyze the TEs and report the complete TE contents.
-
Line 209, homology has no high or low, the genes are either homologous or not.
-
Line 380-387, the authors should talk about what you find in T. coccineum genome, since this is a T.coccineium genome paper.
Author Response
Thank you very much for your constructive suggestions. We have addressed all of the reviewer's comments. The reviewer's comments are displayed in italics. Our responses follow each comment.
To Reviewer #1:
- Phylogenetic analysis of plants. Only ITS1 and ITS2 are not enough to draw a phylogenetic tree, especially the plants selected all have good genome assemblies. It is 2022, the authors should use multiple markers, rather than ITS alone for this type of analysis.
<Response>
As per your comments, we also analyzed other gene markers rbcL and matK (available on the NCBI database). We have added these results to the plant phylogenetic tree as follows:
In the Figures and Figure legends
“Figure 1. (A) Flowers of Tanacetum cinerariifolium and Tanacetum coccineum. (B) The phylogenetic tree of plants in this paper. To generate the phylogenetic tree, the sequences of internal transcribed spacer (ITS)1, ITS2, ribulose-1,5-bisphosphate carboxylase/oxgenase large subunit (rbcL) and maturase K (matK) were obtained from the NCBI database or BLASTN search. The nucleic acid sequences of ITS1, ITS2, rbcL and matK were aligned using CLUSTAL W-mpi 0.13, and maximum-likelihood phylogenetic trees based on a JTT matrix-based model with 1000 bootstraps were created using IQ-TREE 2.0.3 and FigTree v1.4.4 software.” (Line 65-72 in the revised manuscript.)
In the Materials and Methods section
“3.1. Phylogenetic analysis of plants in this paper
The sequences of internal transcribed spacer (ITS)1, ITS2, ribulose-1,5-bisphosphate carboxylase/oxgenase large subunit (rbcL) and maturase K (matK) were obtained from the NCBI database or BLASTN. Accession No.; T. cinerariifolium: AB359720.1 (ITS1), AB359806.1 (ITS2) and MT104464.1 (rbcL and matK); T. coccineum: AB359721.1 (ITS1), AB359807.1 (ITS2) and MT104463.1 (rbcL and matK); A. annua: KC493085.1 (ITS1 and ITS2) and MF623173.1 (rbcL and matK); H. annuus: KF767534.1 (ITS1 and ITS2), L13929.1 (rbcL) and AY215805.1 (matK); N. tabacum: AJ300215.1 (ITS1 and ITS2), AP019625.1 (rbcL) and MZ707522.1 (matK); A. thaliana: X52320.1 (ITS1 and ITS2), NC_000932.1 (rbcL) and MK380721.1 (matK); O. sativa: KM036282.1 (ITS1 and ITS2), D00207.1 (rbcL) and KM103369.1 (matK). The sequences of ITS 1, ITS2, rbcL and matKof C. seticuspe were detected by BLASTN (version 2.7.1) [27] with the sequences of these genes of A. thaliana as queries. The nucleic acid sequences (except O. sativa sequences due to their low similarity to other plants’) were aligned using CLUSTAL W-mpi 0.13 [32], and maximum-likelihood phylogenetic trees based on a JTT matrix-based model [33] with 1000 bootstraps were created using IQ-TREE 2.0.3 [35] and FigTree v1.4.4 software (http://tree.bio.ed.ac.uk/software/figtree/).” (Line 394-410 in the revised manuscript.)
In the Supplemental Materials
“Figure S1. Nucleic acid sequence alignments of internal transcribed spacer (ITS)1, ITS2, rbcL and metK. The sequences of internal transcribed spacer (ITS)1, and ITS2, ribu-lose-1,5-bisphosphate carboxylase/oxgenase large subunit (rbcL) and maturase K (matK) were obtained from the NCBI database or BLASTN. Accession No.; T. cinerariifolium: AB359720.1 (ITS1), AB359806.1 (ITS2) and MT104464.1 (rbcL and matK); T. coccineum: AB359721.1 (ITS1), AB359807.1 (ITS2) and MT104463.1 (rbcL and matK); A. annua: KC493085.1 (ITS1 and ITS2) and MF623173.1 (rbcL and matK); H. annuus: KF767534.1 (ITS1 and ITS2), L13929.1 (rbcL) and AY215805.1 (matK); N. tabacum: AJ300215.1 (ITS1 and ITS2), AP019625.1 (rbcL) and MZ707522.1 (matK); A. thaliana: X52320.1 (ITS1 and ITS2), NC_000932.1 (rbcL) and MK380721.1 (matK); O. sativa: KM036282.1 (ITS1 and ITS2), D00207.1 (rbcL) and KM103369.1 (matK). The sequences of ITS 1, ITS2, rbcL and matK of C. seticuspe were detected by BLASTN (version 2.7.1) with the sequences of these genes of A. thaliana as queries. The nucleic acid sequences (except O. sativa sequences due to their low similarity to other plants’) were aligned using CLUSTAL W-mpi 0.13. Asterisks and hyphens denote conserved nucleic acids and gaps, respectively.”
- Please report the number of Ns for contigs and scaffold assemblies in Table 2. I wonder about the change of gaps after you applied the gap-filling step.
<Response>
Thank you for your helpful comment. As per your suggestions, we have changed Table 2 to clarify the change of gaps from scaffolding to draft genome as follow:
In the Results section 2.3
Table 2
The numbers of Ns were added in Table 2. The column of “scaffolds” was separated into “scaffolds (before gap-filling)” and “draft genome” to clarify the change of gaps through the gap-filling step.
- Did you do repeat masking before gene annotation? You should find a lot smaller gene numbers after masking repeats. I would suggest the authors re-annotate the genome.
<Response>
Thank you for your helpful comment. As per your suggestions, we performed RepeatMasker to mask repeat regions and compared the predicted genes for the unmasked genome and masked genome (please refer the attached files). The comparison verified that RepeatMasker misdetected some highly multiplied genes including TcoLOX1 (Tco_0863779), which is essential for pyrethrin biosynthesis. Consequently, we adopted GyDB- and BLAST-based repeat detection in this paper.
- The number of TEs together with the number of protein-coding genes are 876,474, the authors annotated a total of 1,582,136. What are the other half of the predicted genes?
<Response>
A total of 1,582,136 genes were not annotated genes but predicted genes by using AUGUSTUS. Since the remaining 705,662 genes were not annotated as known protein-coding genes or TEs, they were excluded from the analyses in this study.
- Gypsy (Athila, Tat, Galadriel, Teina, Del/Tekay) and Copia (TAR/Tork, Angela/Tork, Ikeros/Tork, Maximus/Sire, Ivana/Oryco, Ale/Retrofit…) are the two big classes of TEs, most of the TEs in plant genomes are these two superfamilies, it is not necessary to present the subgroups, but lost resolutions for the lower amount TEs. I recommend the authors use RepeatMasker to reanalyze the TEs and report the complete TE contents.
<Response>
Thank you for your comment. As mentioned in No. 3 comment, we performed RepeatMasker to detect TE regions (please refer the attached files). The comparison verified that some of LTRs were not detected by RepeatMasker. Moreover, most RepeatMasker-annotated repeats were LTRs, which were detectable by both Repeatmasker and GyDB. Additionally, non-LTR repeats, which are only detectable by Repeatmasker, were found to be faint in the genome (see “Results only for reviewers” on the last pages). Such a small amount of non-LTR repeats cannot affect the genome size. Furthermore, GyDB was suitable for comparison of the TE contents of T. coccineum with that of other Asteraceae plants, which were calculated using GyDB in the previous study (Yamashiro et al., Sci Rep., 9(1):18249, 2019). Consequently, we concluded that GyDB-based annotation is sufficient for analysis of TEs in this study.
- Line 209, homology has no high or low, the genes are either homologous or not.
<Response>
Thank you for your helpful comment. We have replaced from “T. coccineum predicted proteins with high homology” with “T. coccineum predicted proteins with high sequence similarity”. The revised sentence was as follows:
In the Results
Section 2.5
“T. coccineum predicted proteins with high sequence similarity to 9 known T. cinerariifolium pyrethrin biosynthesis-related proteins (TciADH2 [20], TciALDH1 [20], TciCCH [21], TciCCMT [21], TciCDS [22], TciGLIP [23], TciJMH [24], TciLOX1 [25], TciPYS [26],) were detected by BLASTP [27] (Table 5), indicating that a complete set of known pyrethrin-related enzymes is conserved in the T. coccineum genome.” (Line 210-214 in revised manuscript).
- Line 380-387, the authors should talk about what you find in T. coccineum genome, since this is a T.coccineium genome paper.
<Response>
Thank you for your helpful comment. We have incorporated “Section 4. Conclusions” into Section 2.7 to focus on the argument that the difference of genomic features between T. coccineum and T. cinerariifolium is implicated with the pyrethrin production. We have revised as follows:
In the Results
Section 2.7
“Our previous study implicated a gas-induced histidine kinase in the VOC-mediated regulation of pyrethrin production in T. cinerariifolium [2]. The present study also suggested a correlation between the extent of pyrethrin production and the number of histidine kinase proteins in T. coccineum and T. cinerariifolium. These results suggested that T. coccineum have not acquired a gas (i.e., VOC)-induced pyrethrin production system via species-specific multiplication of histidine kinase-encoding genes distinct from T. cineariifolium. Investigation of the functional relationship between histidine kinases and VOC-induced pyrethrin production is underway.
In this study, we elucidated the draft genome of T. coccineum. Comparative genomic analyses between T. coccineum and a closely related species, T. cinerariifolium, revealed characteristic features of T. coccineum genes, leading to the difference of pyrethrin production between T. coccineum and T. cinerariifolium” (Lines 389-392 in revised manuscript)

Reviewer 2 Report
Actually the manuscript tilted with" Draft genome of Tanacetum coccineum: genomic comparison of 2 closely related Tanacetum-family plants". The article is written in good manner, the idea excellent and the data is satisfied to represent in high quality journal likes yours. The work achieved in this manuscript which compared with the genome of two closely related plants which both of them able to produce insecticidal compounds. But one of them is highly producing of these active components and the other also produces it but in low manner. Actually, the authors did not mention the following:
1- Are the two plants collected from different environment? If they are different in locations it will be normal that the active materials should be differed and the micro RNA genes will follow these changes. Ribosomes also affected with the environmental changes, so, authors should clarify this point to make the manuscript highly cited.
2- Did the authors make analysis for the extracts of the two plants and studying the active components in these extracts both quality and quantity using chromatography analysis. I think if they did biochemical analysis for the two plants it will be good.
3- I do ask the authors to make another study on the ribosome structure and function for the two examined plants, by which we will notice that reibosomes play an important role in the production of the active insecticidal compounds in one plant and do not in the other. This will indicate which this behave was selected and base on what.
4- The introduction should be prolonged and have more recent references
5- Conclusion should be summarized because it is too long.
Author Response
Thank you very much for your suggestions. We have addressed all of the reviewer's suggestions. Comments of the reviewer are displayed in italics. Our responses follow each comment.
To Reviewer #2:
- Are the two plants collected from different environment? If they are different in locations it will be normal that the active materials should be differed and the micro RNA genes will follow these changes. Ribosomes also affected with the environmental changes, so, authors should clarify this point to make the manuscript highly cited.
<Response>
To generate the T. coccineum draft genome, the genome DNA of T. cocccineum was extracted from their commercially available seeds. Analyses of RNA (i.e. RNA-seq) are of interest and will be performed in the future study.
- Did the authors make analysis for the extracts of the two plants and studying the active components in these extracts both quality and quantity using chromatography analysis. I think if they did biochemical analysis for the two plants it will be good.
<Response>
Various secondary metabolites have been reported in several papers regarding T. cinerariifolium. Although T. coccineum was found to contain much fewer amount of pyrethrins than that of T. cinerariifolium as mentioned in the introduction section of this manuscript, little is known about secondary metabolites of T. coccineum. Since this paper, as other papers on genomic analysis, focuses on the genome sequence, this manuscript does not describe metabolites. However, metabolomic analysis of T. coccineum is a challenging theme and expected to be studied in our future study.
- I do ask the authors to make another study on the ribosome structure and function for the two examined plants, by which we will notice that reibosomes play an important role in the production of the active insecticidal compounds in one plant and do not in the other. This will indicate which this behave was selected and base on what.
<Response>
The relationship between the behavior of ribosomes and pyrethrin production has been not reported. Please also see the response to this reviewer’s comment 1.
- The introduction should be prolonged and have more recent references.
<Response>
We confirmed that the introduction of this manuscript includes sufficient recent relevant findings and references. We greatly appreciate if you could kindly suggest recent and suitable references.
- Conclusion should be summarized because it is too long.
<Response>
Thank you for your helpful comment. We have incorporated “Section 4. Conclusions” into the conclusion of section 2.7 to clarify our argument as follows:
In the Results
Section 2.7
“In this study, we elucidated the draft genome of T. coccineum. Comparative genomic analyses between T. coccineum and a closely related species, T. cinerariifolium, revealed characteristic features of T. coccineum genes, leading to the difference of pyrethrin production between T. coccineum and T. cinerariifolium.” (Lines 389-392 in revised manuscript).
Round 2
Reviewer 1 Report
Thank you for the responses, they addressed my comments and confusion clearly.